# Hypergraph Neural Network for Integer Programming with High-Degree Terms

## Abstract

Complex real-world optimization problems often involve not only discrete decisions, but also nonlinear relationships between variables represented in constraints or objectives. A class of such problems can be modeled as integer programming with high-degree terms, such as quadratic integer programming. The nonlinearity makes integer programming problems far more challenging than their linear counterparts. In this paper, we propose a hypergraph neural network (HNN) based method to solve integer programming with high-degree terms. First, we present a high-degree term-aware hypergraph representation to effectively capture both high-degree information and variable-constraint interdependencies. Then, a hypergraph neural network, that integrates convolution between variables and high-degree terms with convolution between variables and constraints, is proposed to predict solution values. Finally, a search process initialized from the predicted solutions is performed to further refine the results. Comprehensive experimental evaluations across a range of benchmarks demonstrate that our method consistently outperforms both learning-based approaches and state-of-the-art solvers, ultimately delivering superior solution quality with favorable efficiency.

## 1 Introduction

Integer programming has been widely applied to real-world applications involving discrete decisions, such as photolithography scheduling (Deenen et al., 2023), supply chain optimization (Bai et al., 2011), and routing optimization (Wu et al., 2022). Many integer programming problems are NP-hard, requiring computational time and memory that grow exponentially with problem size to be solved to optimality. In particular, nonlinear integer programming (NLIP) frequently arises in practice due to physical laws (Ahmadi & Majumdar, 2016), statistical measures (Lejeune & Margot, 2016), nonlinear regression (Seyedan & Mafakheri, 2020), and other complex relationships. The presence of nonlinearity makes these problems even more challenging to solve, highlighting the need for efficient solution methods that go beyond traditional techniques.

Over the past few decades, many algorithms have been proposed to address the challenges of NLIP, typically following two main approaches. Local approaches rely on gradient information to find locally optimal solutions (Bazaraa et al., 2006), but often struggle with complex problem structures containing multiple local optima. Global approaches follow a divide-and-conquer strategy, partitioning the solution space and searching within each partition to identify the optimal solution. Examples include spatial branch-and-bound (Smith & Pantelides, 1999), which recursively partitions the solution space and solves convex relaxations to establish bounds on the original problem, and outer approximation (Kesavan et al., 2004), which iteratively constructs linear approximations of the nonlinear feasible region. Despite their theoretical guarantees in reaching global optimality, global approaches often incur prohibitive computational time for instances with highly nonlinear terms or intricate constraint structures. Furthermore, algorithms for these approaches are typically closed-source or tailored to specific NLIP, which restricts their broader application and potential for improvement. These limitations motivate the exploration of alternative paradigms.

A promising alternative paradigm is machine learning, which has driven major advances in integer linear programming (ILP) recently. The advances include learning better policies within specific solvers, such as branching (Gasse et al., 2019; Nair et al., 2021; Maudet & Danoy, 2025) and presolving (Liu et al., 2024), as well as learning general guidance for ILP solvers, such as solution

prediction (Ding et al., 2020; Geng et al., 2025) and neighborhood selection (Han et al., 2023; Ye et al., 2023). However, due to the fundamental differences between linear and nonlinear formulations, they are not directly applicable to NLIP. This gap underscores the substantial opportunity to develop advanced learning techniques capable of addressing more complex problem classes.

Despite this promise, research on learning-based methods for NLIP remains relatively limited with only a handful of works (Bonami et al., 2022; Ghaddar et al., 2023; Ferber et al., 2023). These methods are mainly built on specific problem structures or algorithms, thus restricting their broader applicability. It highlights the need for more general learning-based methods that can effectively address a wide range of NLIP problems and operate across different solvers.

To address these limitations, this paper aims to push the frontier of learning-for-NLIP towards solving general integer programming with high-degree terms (IPHD), a natural and important subclass of NLIP, such as quadratic and quintic integer programming. By Taylor's formula (Rudin, 1987), IPHD captures many practical nonlinearities and is representative of NLIP challenges that cannot be efficiently solved by current solvers. Instead, we target learning-based NLIP and propose a hypergraph neural network (HNN)-based model that predicts variable values in optimal solutions based on a hypergraph representation of problem instances. The predicted solution serves as an effective initial solution, which can be further refined by any solver or complementary search algorithm. Our major contributions are summarized as follows.

- We develop a hypergraph representation for general integer programming problems with high-degree terms, which encodes interactions among variables within high-degree terms and relations between variables and constraints in the overall problem structure.
- We propose a hypergraph neural network that learns the mapping between problem instances and their corresponding optimal solutions. It applies a convolution across variables and high-degree terms to capture variable representations, together with a convolution between variables and constraints to further involve variable-constraint interdependencies.
- We conduct experiments on diverse benchmark datasets. They demonstrate the superior performance of our method in enhancing Gurobi and SCIP for solving quadratic and quintic integer programming problems with much higher efficiency.

## 2 RELATED WORK

### 2.1 LEARNING-BASED METHODS FOR ILP

This line of research can be broadly categorized into two classes (Bengio et al., 2021; Zhang et al., 2023). The first class concerns learning key policies within solvers. Among these, the most studied include variable selection (Gasse et al., 2019; Gupta et al., 2020; Sun et al., 2021; Nair et al., 2021; Zarpellon et al., 2021; Feng & Yang, 2025; Li et al., 2025) and node selection (Labassi et al., 2022; Maudet & Danoy, 2025). Other policies include cutting plane selection (Deza & Khalil, 2023; Tang et al., 2020; Huang et al., 2022; Wang et al., 2024), primal heuristic selection (Chmiela et al., 2021), parameter tuning (Xu et al., 2011), and presolving settings (Liu et al., 2024; Kuang et al., 2025).

The second class focuses on learning general policies that are applicable across different solvers. One approach involves predicting solutions, either as initial solution values for further refinement (Song et al., 2020; Ding et al., 2020; Huang et al., 2024) or as direct feasible solutions (Geng et al., 2025; Liu et al., 2025; Heydaribeni et al., 2024; Tang & Khalil, 2024). Another direction involves learning strategies in general heuristics, including neighborhood selection strategy for large neighborhood search (Liu et al., 2022; Ye et al., 2023; Han et al., 2023; Huang et al., 2023; Ye et al., 2025; Zhang et al., 2025) and search strategy for diving heuristics (Nair et al., 2021).

### 2.2 LEARNING-BASED METHODS FOR NLIP

The learning-based methods for NLIP remain underexplored in the literature. A few noteworthy contributions have developed learning-for-NLIP methods in specific problems. Bonami et al. (2022) trained a classifier to decide whether linearizing quadratic integer programs leads to better solver performance. Ferber et al. (2023) proposed to learn surrogate linear objective functions for nonlinear programs with linear constraints. Tang et al. (2024) introduced differentiable correction layers for

end-to-end learning on parametric nonlinear programming with fixed problem structure. Chen et al. (2025) studied the theoretical expressive power of graph neural networks for quadratic terms. While these works represent valuable progress, they are all tailored to specific problem settings and do not generalize to integer programming with high-degree terms (IPHD).

Two recent studies are more directly related to our work. Xiong et al. (2024) developed a hypergraph neural network to predict solutions for quadratic programming. However, both their hypergraph representation and neural network are restricted to quadratic terms, whereas our framework is designed to handle IPHD with arbitrary degrees. Ghaddar et al. (2023) applied quantile regression to learn instance-specific branching rules inside a closed-source solver. This approach requires access to internal solver modifications and is limited to a specific solver, while our approach predicts solutions that can be applied as external initial solution values for any solver without internal changes.

Together, these studies highlight the promise and the gap of learning-based methods. Our work fills this gap by proposing a hypergraph neural network framework that addresses integer programming with arbitrary high-degree terms and integrates seamlessly with existing solvers.

## 3 PRELIMINARIES

### 3.1 DEFINITION OF INTEGER PROGRAMMING WITH HIGH-DEGREE TERMS

Integer programming with high-degree terms (IPHD) refers to a class of optimization problems to maximize or minimize an objective function defined over a set of integer variables, while satisfying a set of constraints. In IPHD, either the objective function or the constraints are expressions of linear, quadratic, or higher-order monomial terms. Formally, the mathematical formulation of IPHD with $n$ variables and $m$ constraints is presented as follows for clarity:

$$\min_{x} / \max_{x} \quad \sum_{|\alpha| \leq d_0} c_{0,\alpha} \prod_{i=1}^{n} x_i^{\alpha_i}, \tag{1}$$

$$\text{s.t.} \quad \sum_{|\alpha| \leq d_j} c_{j,\alpha} \prod_{i=1}^{n} x_i^{\alpha_i} \leq b_j, \quad j = 1, 2, \ldots, m, \tag{2}$$

$$l_i \leq x_i \leq u_i, \quad i = 1, 2, \ldots, n, \tag{3}$$

$$x_i \in \mathbb{Z}, \quad i = 1, 2, \ldots, n, \tag{4}$$

where $\alpha = (\alpha_1, \alpha_2, \cdots, \alpha_n) \in (\mathbb{Z}_+ \cup \{0\})^n$ represents the vector of variable degrees; $|\alpha|$ represents the sum of all elements in $\alpha$; $c_{0,\alpha}$ and $c_{j,\alpha}$ denote the coefficients of the term indexed by degree $\alpha$ in the objective function and in the $j$-th constraint, respectively; $d_0$ and $d_j$ are maximum degrees for the objective function and the $j$-th constraint; $b_j$ is the right-hand-side scalar of the $j$-th constraint; $l_i$ and $u_i$ are lower and upper bounds for integer variable $x_i$, respectively.

### 3.2 GRAPH REPRESENTATIONS FOR INTEGER PROGRAMMING

Graph-based representations are commonly used to transform integer programming (IP) instances into structures suitable for graph neural network processing. The seminal work of Gasse et al. (2019) introduced a bipartite graph in which one set of nodes represents variables and the other represents constraints, with edges encoding variable-constraint incidences, i.e., a variable appearing in a constraint. Building on this idea, Ding et al. (2020) extended the representation to a tripartite graph by adding nodes for representing the objective function, thereby enriching the structural information. Subsequent GNN-based representations of IP are often built on bipartite graphs, owing to their effectiveness and simplicity (Gupta et al., 2020; Sun et al., 2021; Nair et al., 2021; Wu et al., 2021; Liu et al., 2022; Labassi et al., 2022; Han et al., 2023; Ye et al., 2023; Huang et al., 2023; Liu et al., 2024; Huang et al., 2024; Liu et al., 2025; Zhang et al., 2025).

While effectively capturing variable-constraint relationships, the current graph-based representations are restricted to pairwise interactions and thus struggle to model the nonlinear or higher-order structures that frequently arise in practical IP problems. To overcome this limitation, Heydaribeni et al. (2024) used hyperedges to connect variables appearing in the same constraint, and Xiong

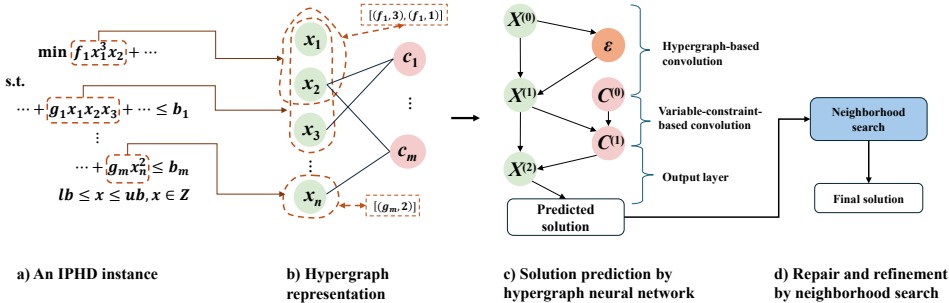

a) An IPHD instance    b) Hypergraph representation    c) Solution prediction by hypergraph neural network    d) Repair and refinement by neighborhood search

Figure 1: **The framework of the proposed method.** For an IPHD problem instance shown in a), our method first transforms it into a hypergraph shown in b), where orange circles denote hyperedges representing high-degree terms. Their raw features (term coefficients and variable degrees) are illustrated in the orange dashed boxes. This hypergraph is then processed by a hypergraph neural network shown in c) for representation learning and solution prediction, where $\epsilon$, $X^{(i)}$, and $C^{(i)}$ represent the embeddings for hyperedges, variables after the $i$-th update, and constraints after the $i$-th update, respectively. Finally, a neighborhood-search based repair-and-refinement process shown in d) turns the predicted results into a high-quality feasible solution.

et al. (2024) employed hyperedges to represent quadratic terms involving both variables and constraints. Nonetheless, both hyperedge-based representations remain limited to specific problems (e.g., IP with linear or quadratic terms) and cannot generalize to IP instances with arbitrary high-degree terms. In this paper, we address this gap by proposing a hypergraph neural network tailored for learning representations of IP with high-degree terms (IPHD).

## 4 METHODOLOGY

This section presents our hypergraph neural network framework for tackling IPHD problems, including hypergraph representation, solution prediction via hypergraph neural network, and solution repair and refinement. The overview of our framework is illustrated in Figure 1 and detailed below.

### 4.1 HIGH-DEGREE TERM-AWARE HYPERGRAPH REPRESENTATION

Representing general IPHDs poses two unique challenges: (i) high-degree terms induce multi-variable interactions that cannot be captured by standard pairwise connections, and (ii) the satisfaction of constraints depends intricately on variable assignments, forming another essential relationship. To address these challenges, we encode an IPHD instance as a hypergraph.

Formally, our hypergraph is defined as $\mathcal{G} = (\mathcal{V}, \mathcal{C}, \mathcal{H}, \mathcal{E})$, where $\mathcal{V}$ denotes variable vertices, $\mathcal{C}$ denotes constraint vertices, $\mathcal{H}$ denotes hyperedges, and $\mathcal{E}$ denotes standard edges. In specific, each variable $x_j$ in an IPHD instance is represented by a vertex $v \in \mathcal{V}$, and each constraint is represented by a vertex $c \in \mathcal{C}$. For every high-degree term $c_\alpha \prod_{i' \in \{1, \cdots, |\mathcal{V}|\}} x_{i'}^{\alpha_{i'}}$ (with $\sum_{i' \in \{1, \cdots, |\mathcal{V}|\}} \alpha_{i'} \geq 2$) appearing in either the objective function or a constraint, we create a hyperedge $\epsilon \in \mathcal{H}$ that connects all variables contained in the term. The raw features of the hyperedge $\epsilon$ are defined as $\{\omega_{v_{i'}\epsilon} = (c_\alpha, \alpha_{i'})\}_{v_{i'} \in \mathcal{N}_\epsilon}$, where $\mathcal{N}_\epsilon$ represents variables contained by $\epsilon$. To model variable-constraint relationships, we add an edge $e_{vc} \in \mathcal{E}$ between variable vertex $v$ and constraint vertex $c$ whenever the corresponding variable appears with a nonzero coefficient in the corresponding constraint. The associated coefficient and variable degree are assigned as features of this edge, ensuring that the numerical dependency between variable and constraint is preserved.

The hypergraph-based representation integrates both structural and parametric information of an IPHD instance, and provides a foundation for the hypergraph neural network introduced in the next subsection. A complete specification of the raw features is provided in Appendix A.1.

For example, Figure 1(b) illustrates the hypergraph representation for the IPHD instance in Figure 1(a). Variable vertices (left) and constraint vertices (right) represent variables and constraints separately; edges (straight lines) represent variable-constraint relationships, connecting a variable to

each constraint in which it appears with a nonzero coefficient (e.g., $x_3$ is connected to $c_1$); hyperedges (circles) capture the complex relationships of variables in high-degree terms such as $f_1 x_1^3 x_2$ and $g_m\, x_n^2$. The term $f_1 x_1^3 x_2$ has coefficient $f_1$ and two variables with exponents 3 and 1, so the raw features of its hyperedge are $\{(f_1, 3), (f_1, 1)\}$. Similarly, the raw features of the term $g_m\, x_n^2$ are $\{(g_m, 2)\}$. Both raw features are illustrated in dashed boxes.

## 4.2 Solution Prediction via Hypergraph Neural Network

According to the hypergraph representation of IPHD instances, the graph neural networks should be able to capture two complementary relationships: (i) high-order interactions among variables induced by high-degree terms, and (ii) interdependencies between variables and constraints. To this end, we first build on concepts from Hypergraph Neural Networks (HNNs), which enable message passing between vertices and hyperedges to model higher-order structures (Kim et al., 2024). With an HNN, we introduce a hyperedge-based convolution that aggregates information from hyperedges and integrates them into variable embeddings. Second, we design a variable-constraint-based convolution that propagates information along standard edges that represent variable-constraint interdependencies. In Section 5.3, an ablation study demonstrates the effectiveness of the above modules. Finally, the variable embeddings are passed through an output layer to generate the predictions of variable values in high-quality solutions. The architecture of our model is illustrated in Figure 1(c).

### 4.2.1 Hyperedge-based Convolution

Our HNN begins with a hyperedge-based convolution, applied to hyperedges and variable vertices contained by them, in order to effectively extract higher-order information arising from the high-degree terms. Inspired by Huang & Yang (2021), we formulate the convolution as presented in Eq. 5 and Eq. 6. This convolution proceeds in two steps: first, hyperedges aggregate embeddings from all variable vertices they directly connect (Eq. 5); then, the updated hyperedge embeddings are propagated back to the associated variable vertices (Eq. 6). In this way, information from high-degree terms is jointly integrated into the embeddings of the variables they contain.

$$h_\epsilon \leftarrow \sum_{v \in \mathcal{N}_\epsilon} h_v h_{v\epsilon}, \forall \epsilon \in \mathcal{H}, \tag{5}$$

$$h_v \leftarrow \phi_\mathcal{H}(h_v, \text{mean}(\{h_\epsilon h_{v\epsilon}\}_{\epsilon \in \mathcal{N}_v})) + h_v, \forall v \in \mathcal{V}, \tag{6}$$

Formally, $h_\epsilon$ and $h_v$ denote the embeddings for hyperedge $\epsilon$ and variable vertex $v$, respectively; $h_{v\epsilon}$ is the embedding obtained from the raw features $\omega_{v\epsilon}$ (see Section 4.1) through a two-layer multi-layer perceptron (MLP). $\mathcal{N}_v$ is the set of hyperedges containing the variable $v$; $\mathcal{N}_\epsilon$ is the set of variable vertices contained in the hyperedge $\epsilon$. The function $\phi_\mathcal{H}$ is parameterized by another two-layer MLP activated by LeakyReLU. The hyperedge-based convolution is repeated for $L$ iterations. The input embeddings to the first iteration are initialized by applying two-layer MLPs to raw features, while those to the latter iterations are from the previous iteration.

### 4.2.2 Variable-Constraint-based Convolution

After the hyperedge-based convolution embeds higher-order relationships into variable vertices, the model still needs to account for variable-constraint interdependencies. To this end, we apply a variable-constraint-based convolution that explicitly processes message passing along the edges connecting variable and constraint vertices.

This convolution operates through bidirectional message passing: variable embeddings (that already aggregate higher-order information from the hyperedge-based convolution) are first propagated to constraint vertices (Eq. 7), and the updated constraint representations are then passed back to the variable vertices (Eq. 8). Through the two-step convolutions, variable embeddings are further informed by the constraints in which the variables connect.

$$h_c \leftarrow f_\mathcal{C}(h_c, \sum_{v \in \mathcal{N}_c} \phi_\mathcal{C}(h_c, h_v, h_{vc})) + h_c, \forall c \in \mathcal{C}, \tag{7}$$

$$h_v \leftarrow f_\mathcal{V}(h_v, \sum_{c \in \mathcal{N}_v} \phi_\mathcal{V}(h_c, h_v, h_{vc})) + h_v, \forall v \in \mathcal{V}, \tag{8}$$

Formally, $h_c$ denotes the embedding for constraint vertex $c$; $h_v$ denotes the embedding for variable vertex $v$; and $h_{vc}$ denotes the embedding of the edge $e_{v,c}$ connecting $v$ and $c$. The set $\mathcal{N}_c$ contains all variable vertices connected to constraint $c$, while $\mathcal{N}_v$ contains all constraint vertices connected to variable $v$. Finally, $\phi_{\mathcal{C}}$, $\phi_{\mathcal{V}}$, $f_{\mathcal{C}}$, and $f_{\mathcal{V}}$ are implemented as two-layer MLPs activated by LeakyReLU. The variable-constraint-based convolution is executed once. The input embeddings $h_v$ are passed from the hyperedge-based convolution while the inputs $h_c$ and $h_{vc}$ are initialized from raw features of constraint vertices and edges via 2-layer MLPs, separately.

### 4.2.3 Solution Prediction and refinement

After hyperedge-based and variable-constraint-based convolutions, the variable embeddings involve both high-order interactions from high-degree terms and the interdependencies between variables and constraints. To generate solution predictions, we feed these embeddings into a two-layer MLP, which outputs a scalar value for each variable representing its prediction.

The entire HNN is trained in a supervised manner using the binary cross-entropy loss: $\mathcal{L}_{\text{BCE}} = -\frac{1}{N} \sum_{i=1}^{N} [y_i \log(\sigma(\hat{y}_i)) + (1 - y_i) \log(1 - \sigma(\hat{y}_i))]$, where $N$ is the number of logits, $y \in \{0,1\}^N$ and $\hat{y} \in \mathbb{R}^N$ represent the ground truth and the predicted logits, $\sigma(\cdot)$ is the sigmoid function.

The predictions produced by our model can provide initial solution values for downstream algorithms and solvers. To make use of them, we adopt a parallel neighborhood optimization framework in (Ye et al., 2023; Xiong et al., 2024) that embeds an off-the-shelf solver to refine the predicted solution during inference, as illustrated in Figure 1(d). Specifically, the framework employs adaptive large neighborhood search: it first repairs the raw predictions into feasible solutions by fixing promising variables to their predicted values while allowing the remaining variables to be reoptimized by a solver such as Gurobi and SCIP, and then further refines these feasible solutions through additional neighborhood search to achieve better objective values.

## 5 Experimental Results

In this section, we conduct comprehensive experiments to demonstrate the effectiveness of our proposed method in solving IPHD instances. We present and discuss the comparative results in Section 5.2, and an ablation study on model architecture in Section 5.3.

### 5.1 Setup

**Benchmarks** Our experiments were conducted on four IPHD benchmarks. The first two are synthetic quadratic integer programming (QIP) benchmarks introduced by Xiong et al. (2024), derived from two NP-hard problems: the Quadratic Multiple Knapsack Problem (QMKP) and the Random Quadratically Constrained Quadratic Program (RandQCP). Each benchmark contains instances at five scales (Mini, 1000, 2000, 5000, 10000), where the first three scales are used for training and all but Mini are included in testing. The third benchmark is a subset of a public quadratic programming benchmark QPLIB (Furini et al., 2019) selected by Xiong et al. (2024). Finally, we introduce a new synthetic quintic integer programming benchmark based on the Capacitated Facility Location Problem with Traffic Congestion (CFLPTC). This dataset includes five instance scales ($50{\times}10$, $50{\times}20$, $150{\times}30$, $200{\times}30$, $500{\times}100$), with the first four scales used for training and the last three for testing. Detailed formulations and dataset descriptions are provided in Appendix B.

**Baselines** We compare our method against a learning-based method tailored for quadratic programming (QP): NeuralQP (Xiong et al., 2024) which introduces a hypergraph neural network to predict solutions for quadratically constrained QPs. Similar to our HNN model, the model of NeuralQP serves as solution predictors and can be combined with exact solvers for repair and refinement (see Section 4.2.3). We adopt SCIP and Gurobi as the exact solvers in this experiment, and we also included them as standalone baselines to provide a comprehensive comparison. Hereafter, we use "ModelName-G" and "ModelName-S" to represent a learning-based method that integrates the "ModelName" model with Gurobi and SCIP, separately.

We also considered a very recent learning-based baseline, GNN_QP (Chen et al., 2025), which primarily investigates the theoretical expressive power of graph neural networks for quadratic terms.

Table 1: Comparison on QMKP datasets in terms of mean and standard deviation of $\text{gap}_\%$. The best results are highlighted in bold and $*$ indicates statistically significant difference to the best results.

| Method | Train | QMKP | | | | Overall |
|---|---|---|---|---|---|---|
| | | 1000 | 2000 | 5000 | 10000 | |
| Gurobi | – | $14.03^*$ | $5.36^*$ | $29.12^*$ | $17.42^*$ | $16.41^*_{\pm 9.06}$ |
| Neural QP-G | Mini | 3.75 | $0.14^*$ | **0.04** | 0.03 | $0.76_{\pm 0.21}$ |
| | 1000 | 4.00 | $0.14^*$ | 0.04 | 0.04 | |
| | 2000 | – | 0.12 | 0.04 | 0.04 | |
| Ours-G | Mini | 4.06 | $0.14^*$ | 0.05 | 0.03 | $\mathbf{0.75}_{\pm 1.94}$ |
| | 1000 | **3.59** | $0.15^*$ | 0.04 | 0.04 | |
| | 2000 | – | **0.09** | 0.04 | **0.03** | |
| SCIP | – | **5.13** | $31.74^*$ | $35.72^*$ | $6.92^*$ | $19.88^*_{\pm 13.15}$ |
| Neural QP-S | Mini | $19.56^*$ | $0.18^*$ | $2.64^*$ | $6.21^*$ | $5.92_{\pm 6.74}$ |
| | 1000 | $19.11^*$ | 0.15 | 2.60 | $5.95^*$ | |
| | 2000 | – | 0.12 | 2.60 | $6.03^*$ | |
| Ours-S | Mini | $18.10^*$ | $0.20^*$ | **2.60** | $6.07^*$ | $\mathbf{5.41}_{\pm 6.35}$ |
| | 1000 | $16.59^*$ | $0.22^*$ | 2.60 | $6.18^*$ | |
| | 2000 | – | **0.12** | 2.61 | **3.13** | |

While this line of work provides valuable theoretical insights, we observed the empirical performance of their suggested model is markedly worse than ours on the two synthetic quadratic benchmarks. For completeness, we report and discuss these results in the Appendix D.

**Metrics** We evaluate performance using the relative primal gap (in percentage), defined as $\text{gap}_\% = |\text{OBJ} - \text{BKS}|/|\text{BKS} + 10^{-10}| \times 100$, where OBJ is the objective value obtained by a method and BKS is the best-known solution of the instance. For QPLIB instances, BKS values are publicly available, while for synthetic datasets we set BKS to the best objective value found across our experiments. Under the same time limit, a lower $\text{gap}_\%$ indicates solutions closer to BKS and thus stronger performance. To assess statistical significance, we apply the Mann–Whitney U test for unpaired data and the Wilcoxon signed-rank test for paired data, both at the 95% confidence level.

**Implementations** During evaluation, all three learning-based methods followed the same procedure: a trained model first generated a solution prediction, which was then improved using the repair-and-refinement strategy (Section 4.2.3) under a fixed time limit. The exact-solver baselines (SCIP and Gurobi) were instead given the same time to solve each instance from scratch. Time budgets varied by benchmark and instance scale: 100, 600, 1800, and 3600 seconds for QMKP and RandQCP instances at scales 1000, 2000, 5000, and 10000, respectively; 100 seconds for QPLIB instances; and 60, 60, and 1000 seconds for CFLPTC instances at scales 150×30, 200×30, and 500×100. Each method was run five times per instance to account for randomness.

For inference, we matched training and testing benchmarks where possible. On the three synthetic benchmarks, models trained on the same benchmark with identical or smaller instance scales were used. For QPLIB, our model was trained on QMKP-1000 instances, which we found structurally closest to QPLIB. To ensure fairness, we compared against two NeuralQP variants: one trained on QMKP-1000 (as with our model) and one trained on the combined QMKP-1000 and RandQCP-1000 datasets, following the original setup in Xiong et al. (2024). For CFLPTC, our model can directly work on this quintic dataset, whereas the QIP-tailored NeuralQP cannot. To enable comparison, we trained NeuralQP on a quadratic reformulation of CFLPTC obtained by introducing auxiliary variables and constraints. The details of this reformulation are provided in Appendix B. Further details on training and repair-and-refinement settings are provided in Appendix C.

## 5.2 COMPARATIVE EXPERIMENTS

The solving performance of our method and baselines on the four benchmarks is presented in Table 1, Table 2, Table 3, and Table 4. Overall, NeuralQP consistently outperforms Gurobi and SCIP across almost all test datasets, particularly on large-scale instances, which highlights the promise of

Table 2: Comparison on RandQCP datasets by mean and standard deviation of $\mathrm{gap}_\%$. The best results are highlighted in bold and $^*$ indicates statistically significant difference.

| Method | Train | RandQCP | | | | Overall |
|---|---|---|---|---|---|---|
| | | 1000 | 2000 | 5000 | 10000 | |
| Gurobi | – | **2.67** | 4.65* | 4.58* | 5.36* | 4.32*$_{\pm 1.09}$ |
| Neural QP-G | Mini | 3.44* | 2.14* | 3.13* | 3.14 | 2.92*$_{\pm 0.67}$ |
| | 1000 | 3.42* | 2.13 | 3.10* | 3.14* | |
| | 2000 | – | 2.15* | 3.13* | 3.15* | |
| Ours-G | Mini | 3.25 | **2.04** | 3.06 | 3.10 | **2.85**$_{\pm 0.66}$ |
| | 1000 | 3.32* | 2.09 | 3.10 | **3.10** | |
| | 2000 | – | 2.08 | **3.06** | 3.11 | |
| SCIP | – | 38.11* | 41.41* | 37.85* | 53.11* | 42.62*$_{\pm 6.39}$ |
| Neural QP-S | Mini | 0.50* | 0.37* | 0.25* | 0.16 | 0.29*$_{\pm 0.15}$ |
| | 1000 | 0.44 | 0.34 | 0.24* | 0.18 | |
| | 2000 | – | 0.32 | 0.24* | 0.19 | |
| Ours-S | Mini | **0.36** | **0.27** | 0.19 | 0.10 | **0.24**$_{\pm 0.14}$ |
| | 1000 | 0.44* | 0.29 | **0.17** | 0.12 | |
| | 2000 | – | 0.32 | 0.19 | **0.09** | |

Table 3: Comparison on QPLIB instances by mean value and standard deviation of $\mathrm{gap}_\%$. "# ID" denotes the index of the test instance in QPLIB. "Mix" denotes NeuralQP trained on QMKP-1000 + RandQCP-1000, and "QMKP" denotes NeuralQP trained only on QMKP-1000. The best results are highlighted in bold and $^*$ indicates statistically significant difference.

| # ID | Gurobi | NeuralQP-G | | Ours-G | # ID | Gurobi | NeuralQP-G | | Ours-G |
|---|---|---|---|---|---|---|---|---|---|
| | | Mix | QMKP | | | | Mix | QMKP | |
| 2067 | **7.67** | 21.46 | 28.10 | 9.37 | 3860 | 48.95 | **3.30** | 15.92 | 14.69 |
| 2085 | 18.85 | 13.17 | 9.96 | **9.60** | 3841 | 26.69 | 7.85 | 9.90 | **6.13** |
| 3752 | 14.09 | **0.63** | 1.70 | 1.30 | 3883 | 8.25 | 1.04 | **0.36** | 0.63 |
| 2036 | 1.89 | **0.57** | 1.57 | 1.10 | 2957 | 2.11 | 1.41 | 2.07 | **0.71** |
| 2022 | 2.37 | **1.20** | 1.80 | 1.58 | 3402 | **2.80** | 6.95 | 7.10 | 6.64 |
| 2017 | 4.87 | 3.16 | 2.49 | **2.27** | 3347 | 0.52 | 0.33 | **0.20** | 0.34 |
| 2315 | 59.75 | **11.25** | 15.92 | 13.49 | 2733 | 1.38 | 0.37 | **0.28** | 0.38 |
| 3584 | 66.75 | **13.58** | 15.04 | 15.03 | 5962 | 16.89 | 10.76 | 15.04 | **7.06** |

| Overall | Gurobi | NeuralQP-G Mix | NeuralQP-G QMKP | Ours-G |
|---|---|---|---|---|
| | 17.74*$_{\pm 21.08}$ | 6.06$_{\pm 6.99}$ | 7.96$_{\pm 7.90}$ | **5.65**$_{\pm 6.02}$ |

learning-based approaches for integer programming. On the QIP benchmarks, our method achieves performance that is at least comparable to, and in several cases better than NeuralQP, and it also achieves superior overall results across each QIP dataset. This observation is noteworthy given that NeuralQP is specifically designed for QIP, whereas our approach is developed for the more general class of IPHD problems with higher-order terms. On the quintic CFLPTC benchmark, our method produces significantly better solutions than both Gurobi and NeuralQP (on quadratic reformulated instances). In addition, the results indicate that our model can generalize to instances of considerably larger scale than seen during training. These results collectively demonstrate the effectiveness and generality of our approach across diverse benchmarks and instance scales.

## 5.3 ABLATION STUDY

To evaluate the impact of the two convolution modules, hyperedge-based convolution (see Section 4.2.1) and variable-constraint-based convolution (see Section 4.2.2), on the learning ability of our model, we conducted an ablation study. Specifically, we introduced two variants for comparison: w/o-HyConv, which retains only the variable-constraint-based convolution, and w/o-VCConv, which retains only the hyperedge-based convolution. A straightforward removal of one convolution would leave the model unable to capture certain relationships; for example, w/o-VCConv cannot represent dependencies between variables and constraints. To address this and ensure fairness, we designed

Table 4: Comparison on CFLPTC datasets by mean and standard deviation of $\text{gap}_\%$. The best results are highlighted in bold and $^*$ indicates statistically significant difference.

| Method | Train | CFLPTC | | | Overall |
|---|---|---|---|---|---|
| | | 150×30 | 200×30 | 500×100 | |
| Gurobi | – | 51.42$^*$ | 52.76$^*$ | 38.66$^*$ | 48.89$^*_{\pm 11.11}$ |
| Neural QP-G | Small | 28.03$^*$ | 35.77$^*$ | 23.82$^*$ | 37.04$^*_{\pm 13.95}$ |
| | Medium | 42.20$^*$ | 57.24$^*$ | 26.47$^*$ | |
| Ours-G | Small | 9.95$^*$ | 8.96 | 2.78 | **6.59**$_{\pm 6.39}$ |
| | Medium | **5.23** | **7.08** | **2.65** | |

alternative representations that allow w/o-HyConv and w/o-VCConv to access all relationships in IPHDs while differing as little as possible from our original hypergraph representation, which are provided in Appendix C. Apart from the absence of one convolution and the modified representation, all other architectural and implementation settings remain identical to our full method.

We trained w/o-HyConv and w/o-VCConv on the 1000-scaled training datasets from the QMKP and RandQCP benchmarks, and subsequently evaluated their performance on the corresponding 1000-scaled and 2000-scaled test datasets. During inference, Gurobi was employed for repair-and-refinement. For evaluation, we used the validation F1-score to measure predictive performance and $\text{gap}_\%$ to measure solving performance. The results, summarized in Table 5, indicate that both the hyperedge-based and variable-constraint-based convolutions are essential to the HNN architecture, as removing either convolution mechanism leads to a noticeable drop in performance.

Table 5: Comparison of our model and ablation baselines in terms of validation F1-score (left) and $\text{gap}_\%$ (right). The best results are highlighted in bold.

| Method | Validation F1-score | | Mean $\text{gap}_\%$ | | | |
|---|---|---|---|---|---|---|
| | QMKP | RandQCP | QMKP | | RandQCP | |
| | | | 1000 | 2000 | 1000 | 2000 |
| w/o-VCConv | 0.73 | 0.74 | 4.62 | 0.15 | 3.39 | 2.15 |
| w/o-HyConv | 0.74 | 0.77 | 4.82 | **0.13** | 3.37 | 2.15 |
| Ours | **0.78** | **0.79** | **3.59** | 0.15 | **3.32** | **2.09** |

# 6 CONCLUSION

This paper introduces a novel hypergraph neural network (HNN) framework for solving integer programming problems with high-degree terms (IPHD). Our approach contributes two key innovations: a high-degree-aware hypergraph representation that effectively captures variable interactions in high-degree terms and variable-constraint interdependencies inherent in IPHD problems, and a hypergraph neural network architecture that integrates hypergraph-based and bipartite-graph-based convolutions to enable accurate solution prediction. Comprehensive experimental evaluation across quadratic and quintic programming problems demonstrates that our method significantly outperforms both state-of-the-art exact solvers and specialized learning-based approaches, establishing its remarkable effectiveness and practical value for challenging IPHD applications.

While promising, our work represents just one step toward addressing the broader challenges of nonlinear integer programming. Future research directions include: 1) designing more comprehensive representations for general nonlinear instances such as those with trigonometric and logarithmic functions; 2) exploring end-to-end frameworks that directly output feasible solutions without requiring repair mechanisms; and 3) integrating advanced large language models to automate problem solving and reduce reliance on domain-specific knowledge or manual intervention.

## REPRODUCIBILITY STATEMENT

We provide detailed information to ensure the reproducibility of our results. A complete description of our method is given in Section 4.2. Components that are only briefly introduced in the main text, including raw feature selection and the repair-and-refinement procedure, are further detailed in Appendix A.1 and Appendix A.2, respectively. Implementation details for model training are provided in Appendix C, while the evaluation setup is described in Section 5.1. A detailed description of the synthetic benchmarks is included in Appendix B. To further support reproducibility, we will release our code and data on GitHub upon acceptance, enabling researchers to fully replicate our experiments and results.

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

## A  DETAILS OF HNN-BASED FRAMEWORK

We present two key details of the HNN-based framework that were not covered in Section 4, allowing interested readers to reproduce our work.

## A.1 Raw Features of Hypergraph Representation

We present the raw features of our hypergraph representation in Table 6. The table organizes four key components of our hypergraph representation that participate in convolutions. Each row corresponds to one component, with the first column identifying the component name, the second column listing its raw features, and the third column providing detailed descriptions of these features. Specifically, the variable vertices $\mathcal{V}$ are assigned nine-dimensional raw features that encode variable types, bound information, and their roles in the objective function. Constraint vertices $\mathcal{C}$ are assigned four-dimensional raw features based on their constraint sense and right-hand-side values. Hyperedges $\mathcal{H}$ are assigned raw features whose length varies according to the number of variables they contain, as introduced in Section 4.1. For a variable $v$ contained in a hyperedge $\epsilon$, a $\omega_{v\epsilon}$ containing the term coefficient and the variable's exponent is added to $\epsilon$'s raw features. Finally, standard edges $\mathcal{E}$ are assigned two-dimensional features that reflect coefficients and degrees of the corresponding variables within their associated constraints.

Table 6: Raw Features of High-Degree Term-Aware Hypergraph Representation

| Tensor | Feature | Description |
|---|---|---|
| $\mathcal{V}$ | type | (continuous, binary, integer) as a one-hot encoding |
| | lb | Lower bound value of the variable |
| | up | Upper bound value of the variable |
| | inf_lb | Binary indicator (1 if the lower bound is negative infinity, 0 otherwise) |
| | inf_ub | Binary indicator (1 if the upper bound is positive infinity, 0 otherwise) |
| | avg_obj_coe | Average value of coefficients associated with this variable in the objective function |
| | avg_obj_deg | Average degree of this variable across all terms in the objective function |
| $\mathcal{C}$ | sense | $(<, >, =)$ as a one-hot encoding |
| | rhs | Numerical value on the right-hand side of the constraint |
| $\omega_{v\epsilon}$ | deg | Degree of each variable in the high-degree term |
| | coe | Coefficient value associated with the high-degree term |
| $\mathcal{E}$ | avg_coe | Average value of coefficients across all terms containing the variable in the associated constraint |
| | avg_deg | Average degree of this variable across all terms containing it in the associated constraint |

## A.2 Neighborhood Search for Repair-and-Refinement

We implement parallel neighborhood optimization as described in (Ye et al., 2023; Xiong et al., 2024), which incorporates two key components: a Q-repair-based repair strategy that efficiently repairs model predictions into feasible solutions, and an iterated multi-neighborhood search that refines these solutions to achieve higher quality. In the following, we provide detailed descriptions of both components.

### A.2.1 Q-Repair-Based Repair Strategy

The Q-repair begins by selecting the $\alpha n$ variables with the largest predicted loss values to optimize, while fixing the remaining $(1 - \alpha)n$ variables to their predicted values. Here, $\alpha \in [0, 1]$ is a proportion that determines the neighborhood search size and $n$ represents the total number of variables. Then Q-repair traverses constraints to identify those that cannot be satisfied. This identification follows a greedy approach: calculating the upper and lower bounds of each term in the left-hand side, summing these bounds, and comparing the result with the right-hand side. When an unsatisfied constraint is detected, the variables involved in this constraint are incrementally added to the neighborhood until either all variables from that constraint have been incorporated or the neighborhood

reaches a size limit of $\alpha_{\text{ub}}n$ variables. Q-repair terminates after evaluating all constraints and returns the neighborhood (i.e., variables to be optimized) for repair.

Subsequently, the repair strategy employs exact solvers (such as Gurobi and SCIP) to optimize the subproblem defined by the Q-repair neighborhood. If no feasible solution is identified within the allocated time, Q-repair is repeated with an enlarged initial $\alpha = \alpha_{\text{step}} + \text{len}(\text{neighborhood})/n$, followed by another neighborhood search on the new expanded neighborhood. This iterative process continues until a feasible solution is found, or $\alpha$ exceeds 1, or the maximum time to repair-and-refine has been reached.

### A.2.2 Iterated Multi-Neighborhood Search

The iterated multi-neighborhood search begins by generating a set of initial neighborhoods using a sequential filling approach. Specifically, this process first randomly shuffles all constraints. Then, it iteratively processes each constraint by sequentially adding its variables to the current neighborhood. When the predefined neighborhood size limit is reached, a new neighborhood is created and the process continues, until all constraints and their associated variables have been assigned to neighborhoods. This process creates multiple neighborhoods where variables from the same constraint tend to appear together in the same neighborhoods, thereby reducing the likelihood of constraint violations. Next, using the solution obtained by the repair strategy as a starting point, subproblems are formulated based on each neighborhood and optimized using exact solvers.

After that, the algorithm generates crossover neighborhoods to explore combinations of different subproblem solutions. It groups all neighborhoods into pairs. For two neighborhoods $N_1$ and $N_2$ in a pair with their respective subproblem solutions $x^1, x^2$, assuming $x^1$ has equal or better objective value than $x^2$, a crossover neighborhood is created through two steps: 1) constructing a crossover solution $x'$ by taking $x'_i = x^1_i$ for variables in $N_1$ and $x'_i = x^2_i$ for other variables, and 2) applying Q-repair on $x'$. Then, subproblems based on these crossover neighborhoods are optimized. The algorithm selects the best solution among all the candidates, both initial neighborhoods and crossover neighborhoods, to serve as the starting point for the next iteration. These two processes repeat until the predetermined time limit is reached, with the best solution found across all iterations returned as the final result.

## B Details of Benchmarks

This section introduces the details of the synthetic datasets used in our experiments.

### B.1 Details of Synthetic Quadratic Instances

In Section 5.3, we evaluate the efficiency of our HNN-based framework using two synthetic quadratic datasets: QMKP and RandQCP, which are generated and provided by (Xiong et al., 2024). The formulations of these problems are presented below.

The Quadratic Multiple Knapsack Problem (QMKP) extends the classic knapsack problem by incorporating multiple weight constraints and quadratic profit terms. It involves selecting items to place in a knapsack with limited capacity across multiple weight dimensions. Each item yields an individual profit, while specific pairs of items generate additional interactive profits when selected together. The objective is to maximize the total profit while adhering to capacity constraints. QMKP can be formulated as a quadratic programming problem as shown in Eq. 9-11:

$$\max \quad \sum_i c_i x_i + \sum_{(i,j) \in E} q_{ij} x_i x_j, \tag{9}$$

$$\text{s.t.} \quad a_i^k x_i \leq b^k, \quad \forall k \in M, \tag{10}$$

$$x_i \in \{0, 1\}, \quad \forall i \in N, \tag{11}$$

where $x_i$ is a binary variable indicating whether item $i$ is selected, $c_i$ represents the individual profit for item $i$, and $q_{ij}$ denotes the interactive profit obtained by selecting both items $i$ and $j$. The set $E$ contains item pairs with interactive profits, $a_i^k$ represents the $k$-th weight of item $i$, and $b^k$ denotes the knapsack's capacity on the $k$-th weight dimension. $M$ and $N$ represent the total number of weight dimensions and items, respectively.

The Random Quadratically Constrained Quadratic Program (RandQCP) is an extension of the independent set problem. It aims to select vertices from a hypergraph to maximize total weights while satisfying specified constraints on each hyperedge. The quadratic programming formulation of RandQCP is given in Eq. 12-14.

$$\max \quad \sum_{i \in V} c_i x_i, \tag{12}$$

$$\text{s.t.} \quad \sum_{i \in e} a_i x_i + \sum_{i,j \in e, i \neq j} q_{ij} x_i x_j - |e| \leq 0, \quad \forall e \in \mathcal{E}, \tag{13}$$

$$x_i \in \{0, 1\}, \quad \forall i \in V, \tag{14}$$

where $V$ represents the set of vertices, $\mathcal{E}$ denotes the hyperedge set, $c_i$ is the weight associated with vertex $i$, and $a_i$ and $q_{ij}$ are the limitation coefficients for selecting vertex $i$ and vertex pair $(i, j)$, respectively. The term $e$ refers to a specific hyperedge, and $|e|$ indicates the number of vertices contained within hyperedge $e$.

For details of generation and access to the generated datasets, please refer to the original paper by (Xiong et al., 2024).

### B.2 DETAILS OF SYNTHETIC QUINTIC INSTANCES

To evaluate the effectiveness of our HNN-based method on more complex integer programming problems, we generated synthetic quintic datasets based on the Capacitated Facility Location Problem under Traffic Congestion (CFLPTC) inspired by Bai et al. (2011) and Holmberg et al. (1999). The formulation and generation procedures are detailed below.

### B.2.1 FORMULATION OF CFLPTC

CFLPTC extends the standard capacitated facility location problem by incorporating traffic congestion effects. Consider a scenario with $m$ customers $J = \{1, \cdots, m\}$ and $n$ potential facility locations $I = \{1, \cdots, n\}$. Each customer $j$ has a demand $D_j$, while each facility at location $i$ incurs an opening cost $o_i$ and has a capacity $C_i$. Once opened, a facility can serve customers provided that the total demand it satisfies does not exceed its capacity. Each customer must be served by exactly one opened facility. The transportation cost for serving customer $j$ from facility $i$ depends on the distance between them $d_{ij}$ and the traffic congestion level. The objective is to determine which facilities to open and how to assign customers to these facilities, so that the total cost comprising facility opening costs and transportation expenses is minimized. The mathematical formulation is presented in Eq. 15-20.

$$\min \quad \sum_{i \in I} o_i y_i + \sum_{i \in I} \sum_{j \in J} \alpha (1 + 0.15 e_i^{\beta}) d_{ij} x_{ij} \tag{15}$$

$$\text{s.t.} \quad \sum_i x_{ij} = 1, \forall j \in J, \tag{16}$$

$$x_{ij} \leq y_i, \forall i \in I, j \in J, \tag{17}$$

$$\sum_j D_j x_{ij} \leq C_i y_i, \forall i \in I, \tag{18}$$

$$e_i = \frac{\sum_j D_j x_{ij} + b_i}{T_i}, \forall i \in I, \tag{19}$$

$$x_{ij}, y_i \in \{0, 1\}, \forall i \in I, j \in J. \tag{20}$$

where $y_i$ and $x_{ij}$ are binary variables to determine whether to open the facility at location $i$ and whether to assign customer $j$ to the facility at location $i$, separately.

In the objective function Eq. 15, the transportation cost from facility $i$ to customer $j$ is expressed as $\alpha(1 + 0.15 e_i^{\beta}) d_{ij} x_{ij}$, where the term $\alpha(1 + 0.15 e_i^{\beta})$ quantifies the additional cost induced by traffic congestion. This formulation, together with Eq. 19 which determines $e_i$, is derived from the Bureau

of Public Roads (BPR) function, an empirical formula for estimating increased transportation time corresponding to congestion level (United States Bureau of Public Roads, 1964). In this context, $T_i$ represents the total traffic capacity surrounding facility location $i$ and $b_i$ denotes the background traffic flow in the vicinity. The parameters $\alpha$ and $\beta$ are typically set to 1 and 4 respectively, which make CFLPTC a quintic programming problem.

While CFLPTC technically falls under the category of mixed-integer programming due to its combination of binary variables ($x_{ij}$, $y_i$) and continuous variables ($e_i$), it remains essentially an integer programming problem. This is because the continuous variables $e_i$ are merely auxiliary and completely determined by the binary assignment variables $x_{ij}$. Therefore, it is methodologically reasonable to include CFLPTC as a dataset in this work, which focuses on integer programming problems.

### B.2.2 QUADRATIC REFORMULATION OF CFLPTC

In Section 5.1, we compared our method against NeuralQP on the quintic CFLPTC instances. However, NeuralQP is designed exclusively for quadratic optimization problems and cannot directly handle the quintic terms present in the original CFLPTC formulation. To enable this comparison, we reformulated the quintic CFLPTC instances into equivalent quadratic problems by introducing auxiliary variables that decompose higher-order terms. The reformulation strategy systematically replaces quintic terms with chains of quadratic relationships. Specifically, for each $i \in I$, we define $e_{1i} = e_i^2$ and $e_{2i} = e_{1i}^2$, which transform the quintic terms $e_i^4 x_{ij}$ into quadratic terms $e_{2i} x_{ij}$. The complete quadratic reformulation is presented in Eq. 21-24.

$$\min \quad \sum_{i \in I} o_i y_i + \sum_{i \in I} \sum_{j \in J} \alpha(1 + 0.15 e_{2i}) d_{ij} x_{ij} \tag{21}$$

$$\text{s.t.} \quad \text{Eq. 16 - 20,} \tag{22}$$

$$e_{1i} = e_i^2, \forall i \in I, \tag{23}$$

$$e_{2i} = e_{1i}^2, \forall i \in I, \tag{24}$$

It is important to note that while lower-degree objective functions and constraints are generally more tractable for optimization algorithms than their higher-degree counterparts, the reformulation process inevitably introduces additional variables and constraints that can impose significant computational overhead. For CFLPTC instances, the quadratic reformulation requires $2n$ additional variables ($e_{1i}, e_{2i}$) and $2n$ additional quadratic constraints (Eq. 23 and 24), substantially increasing the complexity. The increase of complexity may offset or even outweigh the computational benefits gained from degree reduction, as solvers must now handle a larger search space and a more complicated constraint set. Consequently, reformulating high-degree problems into lower-degree equivalents does not guarantee improved optimization efficiency; the net effect depends on the trade-off between reduced degree and increased problem complexity, which varies with specific problem characteristics and solver capabilities. This trade-off underscores the importance of developing optimization methods that can directly handle high-degree integer programming problems rather than relying solely on quadratic reformulations.

### B.2.3 INSTANCE GENERATION

Following the approach in (Holmberg et al., 1999), we generated datasets at four distinct scales for training, as detailed in Table 7. The notation $U(a, b)$ indicates that the corresponding parameters are randomly sampled from a uniform distribution ranging from $a$ to $b$ (inclusive). Both customer and facility locations were generated within a two-dimensional Euclidean space according to the "Coordinate" specifications in Table 7, with distances calculated using the Euclidean metric. Consistently across all datasets, the total traffic capacity $T_i$ was generated as $U(1, 4) \cdot C_i$, while the background traffic flow $b_i$ was set to $U(0.1, 1) \cdot T_i$.

For testing purposes, we generated 16 instances each at the $150 \times 30$ scale and the $200 \times 30$ scale, adhering to the same parameter settings used for training datasets 3 and 4, respectively. Additionally, we created 10 larger instances at the $500 \times 100$ scale, following the parameter settings of training dataset 1 but with adjusted values for $m$ and $n$. These testing datasets enable comprehensive

Table 7: Setting for CFLPTC Training Dataset Generation

| Dataset | Number | $m$ | $n$ | Coordinate | $D_j$ | $o_i$ | $C_i$ |
|---------|--------|-----|-----|------------|-------|-------|-------|
| 1 | 1605 | 50 | 10 | $U(10, 200)$ | $U(10, 50)$ | $U(300, 700)$ | $U(100, 500)$ |
| 2 | 1119 | 50 | 20 | $U(10, 200)$ | $U(30, 80)$ | $U(300, 700)$ | $U(100, 500)$ |
| 3 | 984 | 150 | 30 | $U(10, 300)$ | $U(10, 50)$ | $U(300, 700)$ | $U(200, 600)$ |
| 4 | 200 | 200 | 30 | $U(10, 200)$ | $U(10, 50)$ | $U(500, 1500)$ | $U(500, 800)$ |

evaluation of our model's capability to effectively tackle complex, large-scale integer programming problems with high-degree terms.

## C  IMPLEMENTATION DETAILS

**Model Details**   First, all raw features of the input hypergraph were transformed into initial embeddings through 2-layer MLPs activated by LeakyReLU, where the dimensions of hidden spaces and output features are 64 and 16, respectively. The number of iterations for executing hyperedge-based convolution is $L = 6$. The negative slopes of all LeakyReLU activations are set to 0.1.

**Training Details**   We utilized AdamW with a learning rate of 1e-4 and weight decay of 1e-4 as the optimizer to train our model. We set the batch size to 64 and training epochs to 100. On each training dataset, our HNN models were trained on a supercomputer node with an NVIDIA A100 GPU and an 18-core Intel Xeon Platinum 8360Y CPU. For fair comparison, we used the same device to train the models of NeuralQP and GNN_QP, with the same hyper-parameter settings as in their original papers.

**Inference Details**   Inference testing was conducted on a personal computer equipped with an 8-core AMD Ryzen 7 7840HS CPU without GPU acceleration. We used Gurobi 12.0.0 and SCIP 9.2.0, the latest versions of both solvers at the time of evaluation.

We implemented the repair-and-refinement algorithm (see Appendix A.2) following the parameter settings proposed by Xiong et al. (2024). Specifically, for the Q-repair-based repair strategy, we initialized the parameter $\alpha$ at 0.1, with $\alpha_{\mathrm{ub}} = 1$ and $\alpha_{\mathrm{step}} = 0.05$. For the iterated multi-neighborhood search, the neighborhood size is defined as half the number of problem variables. For each subproblem occurring in both the Q-repair-based repair strategy and the iterated multi-neighborhood search, we set a maximum wall-clock time of 60 seconds when addressing largest-scale instances: 10,000-scale QMKP and RandQCP problems, and 500×100-scale CFLPTC datasets. All other testing datasets were limited to 30 seconds per subproblem. The repair-and-refinement stops when the total wall-clock time reaches the preset limit (see Section 5.1).

**Details of the Ablation Baselines**   In the ablation studies (Section 5.3), we construct baselines that remain as comparable as possible to our HNN model while omitting the targeted convolution modules. Since simply removing a component would disable the model from capturing one key relationship in IPHD, we make slight but necessary adjustments to their input representations. For w/o-HyConv, the only change is the removal of hyperedges from the representation. For w/o-VCConv, its hypergraph representation contains the same variable and constraint vertices as in our representation but differs in that it has no edges and uses alternative hyperedges. These hyperedges encode both variable interactions in high-degree terms and variable-constraint interdependencies: each term is represented by a hyperedge connecting its variables and the constraint it belongs to. The hyperedge features follow the same design as our representation.

## D  ADDITIONAL EXPERIMENTS TO COMPARE WITH GNN_QP

As stated in Section 5.1, we compared our method with a very recent learning-based baseline, GNN_QP (Chen et al., 2025), which primarily investigates the theoretical expressive power of graph neural networks for quadratic terms. We trained GNN_QP on the two synthetic quadratic benchmarks, QMKP and RandQCP, and evaluated it with Gurobi as repair-and-refinement on the same

Table 8: Comparison on QMKP datasets in terms of mean and standard deviation of $\text{gap}_\%$. The best results are highlighted in bold and $^*$ indicates statistically significant difference to the best results.

| Method | Train | QMKP | | | | Overall |
|---|---|---|---|---|---|---|
| | | 1000 | 2000 | 5000 | 10000 | |
| Gurobi | – | 14.03* | 5.36* | 29.12* | 17.42* | $16.41^*_{\pm 9.06}$ |
| Neural QP-G | Mini | 3.75 | 0.14* | **0.04** | 0.03 | $0.76_{\pm 0.21}$ |
| | 1000 | 4.00 | 0.14* | 0.04 | 0.04 | |
| | 2000 | – | 0.12 | 0.04 | 0.04 | |
| GNN _QP-G | Mini | 12.46* | 0.13 | 0.05 | 0.04 | $2.52^*_{\pm 5.33}$ |
| | 1000 | 5.65 | 0.12 | 0.05 | 0.04 | |
| | 2000 | – | 0.18* | 0.05 | 0.03 | |
| Ours-G | Mini | 4.06 | 0.14* | 0.05 | 0.03 | $\mathbf{0.75}_{\pm 1.94}$ |
| | 1000 | **3.59** | 0.15* | 0.04 | 0.04 | |
| | 2000 | – | **0.09** | 0.04 | **0.03** | |

Table 9: Comparison on RandQCP datasets by mean and standard deviation of $\text{gap}_\%$. The best results are highlighted in bold and $^*$ indicates statistically significant difference.

| Method | Train | RandQCP | | | | Overall |
|---|---|---|---|---|---|---|
| | | 1000 | 2000 | 5000 | 10000 | |
| Gurobi | – | **2.67** | 4.65* | 4.58* | 5.36* | $4.32^*_{\pm 1.09}$ |
| Neural QP-G | Mini | 3.44* | 2.14* | 3.13* | 3.14 | $2.92^*_{\pm 0.67}$ |
| | 1000 | 3.42* | 2.13 | 3.10* | 3.14* | |
| | 2000 | – | 2.15* | 3.13* | 3.15* | |
| GNN _QP-G | Mini | 3.47* | 2.22* | 3.23* | 3.20* | $2.87^*_{\pm 0.80}$ |
| | 1000 | 3.49* | 2.19* | 3.19* | 3.21* | |
| | 2000 | – | 2.18* | 3.13* | 3.22* | |
| Ours-G | Mini | 3.25 | **2.04** | 3.06 | 3.10 | $\mathbf{2.85}_{\pm 0.66}$ |
| | 1000 | 3.32* | 2.09 | 3.10 | **3.10** | |
| | 2000 | – | 2.08 | **3.06** | 3.11 | |

benchmarks. Both training and evaluation used the same implementation settings as in our main experiments. The results, presented in Table 8 and Table 9, demonstrate that our method consistently outperforms GNN_QP on most testing datasets.

# E  ADDITIONAL EXPERIMENTS TO EVALUATE MODEL PREDICTION

In Section 5 we have demonstrated the effectiveness of the complete HNN-based framework composed of both HNN prediction and repair-and-refinement. To assess the quality of our HNN model's predictions as initial solution values without refinement, in this section we conducted additional experiments that isolate the model's predictive performance from the overall framework. We applied our HNN models trained on RandQCP's training data to the RandQCP test sets with 10,000-scaled instances, and models trained on QMKP's training data to the QMKP test sets with 10,000-scaled instances. These largest-scale testing datasets are selected to rigorously assess prediction performance for challenging instances. Since our HNN model generates initial solution values rather than directly producing feasible solutions, we applied the Q-Repair-Based Repair Strategy based on Gurobi (detailed in Appendix A.2) to convert model predictions into feasible solutions, with no further refinement performed. We compared against NeuralQP with identical settings and Gurobi configured to prioritize finding feasible solutions by setting "Params.MIPFocus = 1", "Params.NonConvex = 2", and "Params.SolutionLimit = 1".

We evaluated performance using three comprehensive metrics listed below, and present the comparative results in Table 10.

- Feasible ratio: The percentage of model predictions that yield feasible solutions before repair. A higher feasible ratio indicates stronger constraint satisfaction capability.

- $\text{gap}_\%$: introduced in Section 5.1.
- Wall-clock time: For our method and NeuralQP, it is the time required to obtain a feasible solution through the repair process, while for Gurobi it is the time required to obtain the first feasible solution. Shorter times indicate that the model's predictions can be more efficiently converted into feasible solutions.

The results in Table 10 demonstrate that our HNN model achieves superior solution quality, as evidenced by consistently lower mean $\text{gap}_\%$ values compared to both baselines. This indicates that our model's predictions, after repair, are closer to the best-known solutions and provide higher-quality initial solution values for optimization.

Table 10 also exhibits that our method shows a lower feasible ratio before repair and longer repair times compared to the baseline methods. While these metrics might initially suggest limitations, a closer examination reveals that they do not represent true disadvantages. In terms of feasible ratio, although NeuralQP achieved a higher feasible ratio, both NeuralQP and Gurobi frequently generated trivial solutions with all variables set to zero. Such trivial solutions, while technically feasible, provide less guidance for subsequent refinement processes. Regarding computational time, although our method requires longer repair times than NeuralQP and Gurobi, the actual repair time remains very short (less than 1 second), which is highly acceptable given that 10,000-variable instances typically require extensive search times. In summary, the comparative results demonstrate that our HNN model is a practical choice for generating high-quality initial solution values.

Table 10: Comparison of our HNN model, NeuralQP and Gurobi in terms of prediction performance.

| Method | QMKP | | | RandQCP | | |
|--------|------|------|------|---------|------|------|
| | feasible ratio (%) | $\text{gap}_\%$ | time (ms) | feasible ratio (%) | $\text{gap}_\%$ | time (ms) |
| Gurobi | – | 100 | 5.30 | – | 100 | 2.49 |
| NeuralQP | 100 | 99.10 | 163 | 0 | 53.30 | 392 |
| Ours | 66.67 | 77.40 | 946 | 0 | 51.74 | 835 |

## F  COMPLEXITY ANALYSIS

This section analyzes the memory requirements of the proposed hypergraph representation and the arithmetic time complexity of the proposed HNN's inference. We consider an IPHD instance with $n$ variables, $m$ constraints, and $n_h$ high-degree terms. Let $s$ denote the total number of variable occurrences across all high-degree terms, and let $n_e$ denote the total number of variable-constraint incidences, i.e., the number of times any variable appears with a nonzero coefficient in any constraint. These parameters allow us to demonstrate the efficiency of our method in terms of both memory usage and computational complexity, as shown in the following subsections.

### F.1  MEMORY REQUIREMENT FOR THE HYPERGRAPH REPRESENTATION

According to Section 4.1 and Appendix A.1, hypergraph representation of the IPHD instance comprises four components:

- $n$ variable vertices, each with 9 raw features;
- $m$ constraint vertices, each with 4 features;
- $n_h$ hyperedges, with $s$ vertex-hyperedge coefficients, where each coefficient contains 2 floats;
- $n_e$ edges, each with 2 features;

Variable vertices and constraint vertices can be stored using their indices, while hyperedges and edges can be stored using tuples of vertex indices they contain. In total, hypergraph structure requires $(n + m + s + 2n_e)$ indices to represent. Additionally, there are $(9n + 4m + 2n_e + 2s)$ raw features.

Assuming all indices are stored as 4-byte integers and raw features are stored as 8-byte floats (double precision), the total memory requirement for the hypergraph representation is:

$$\mathbf{bytes} = 76n + 36m + 20s + 24n_e. \tag{25}$$

To illustrate this with a concrete example, consider the largest CFLPTC instances we tested, which involve 500 customers and 100 facilities. As detailed in Section B.2.1, these instances have $n = 50,200, m = 50,700, n_e = 200,300, s = 100,000$. Applying Eq. 25, the total memory requirement is 12,447,600 bytes, or approximately 11.87 megabytes (MB). This represents a very manageable memory overhead for modern hardware, demonstrating that our hypergraph representation remains practical even for large-scale instances.

### F.2 ARITHMETIC TIME COMPLEXITY FOR THE HNN

In this subsection, we analyze the arithmetic complexity of our HNN model during inference. Let $n_{\text{hid}}$ denote the largest dimension among raw features, hidden embeddings, and outputs, and assume we perform $L_{\text{hyper}}$ hypergraph-based convolutions and $L_{\text{bi}}$ bipartite-graph-based convolutions. The complexity analysis for each component is as follows:

- Initial embedding: it is a 2-layer MLP applied on all raw features, with arithmetic complexity $O((n + m + s + n_e)n_{\text{hid}}^2)$;
- Hypergraph-based convolution:
  - Eq. 5 performs weighted summation with complexity $O(sn_{\text{hid}})$;
  - Eq. 6 combines weighted means, a 2-layer MLP, and a residual connection, with complexity $O(sn_{\text{hid}})$, $O(nn_{\text{hid}}^2)$, and $O(nn_{\text{hid}})$, separately. The total complexity is $O(sn_{\text{hid}} + nn_{\text{hid}}^2)$;
  - Overall complexity: $O(L_{\text{hyper}}(sn_{\text{hid}} + nn_{\text{hid}}^2))$;
- Bipartite-graph-based convolution:
  - Eq. 7 combines summations, a 2-layer MLP, and residual connection, with complexity $O(n_e n_{\text{hid}}$, $O(mn_{\text{hid}}^2)$, and $O(mn_{\text{hid}})$, separately. The total complexity is $O(n_e n_{\text{hid}} + mn_{\text{hid}}^2)$;
  - Eq. 8 has similar structure to Eq. 7, with complexity $O(n_e n_{\text{hid}} + nn_{\text{hid}}^2)$;
  - Overall complexity: $O(L_{\text{bi}}(n_e n_{\text{hid}} + mn_{\text{hid}}^2 + nn_{\text{hid}}^2))$;
- Output layer: A 2-layer MLP applied to variable embeddings, with complexity $O(nn_{\text{hid}}^2)$.

Therefore, the overall arithmetic complexity of HNN inference is $O(n_{\text{hid}}(L_{\text{hyper}}s + L_{\text{bi}}n_e) + n_{\text{hid}}^2(L_{\text{hyper}}n + L_{\text{bi}}n + L_{\text{bi}}m))$. Since $n_{\text{hid}}$, $L_{\text{hyper}}$, and $L_{\text{bi}}$ are fixed constants in our experiments (see Section 5.2), the arithmetic complexity simplifies to $O(n + m + s + n_e)$, which scales linearly with the number of variables, constraints, hyperedge density, and edge density.

Hypergraph representations for integer programming problems are typically sparse in both hyperedges and edges, making our HNN model highly efficient. To demonstrate robustness, we consider the extreme case of a fully dense hypergraph representation where every pair of variable and constraint vertices is connected by edges, and all variable vertices are connected within each hyperedge. In this scenario, $s = n_h n$ and $n_e = nm$, yielding a quadratic complexity $O(n(m + n_e))$. This analysis shows that even in such extreme cases, which rarely occur in practice, our HNN model maintains good computational efficiency for inference.

## G LICENSE DESCRIPTION

The licenses and resources of the code, software, and datasets used in this paper are listed in Table 11.

## H STATEMENT OF THE USE OF LARGE LANGUAGE MODELS

Large language models (LLMs) were used solely for writing assistance and language polishing, including grammar correction, sentence restructuring, and clarity improvements. LLMs were not

Table 11: List of licenses for the codes, software and datasets used in this work.

| Resource | Type | Link | License |
|---|---|---|---|
| Gurobi | Software | `https://www.gurobi.com/` | Academic License |
| SCIP | Software | `https://scipopt.org/#scipoptsuite` | Apache 2.0 License |
| AMPL | Software | `https://ampl.com/` | Academic License |
| NeuralQP (Xiong et al., 2024) | Code, Dataset | `https://anonymous.4open.science/r/NeuralQP-Anonymous-7243/` | MIT License |
| QPLIB (Furini et al., 2019) | Dataset | `https://qplib.zib.de/` | CC-BY 4.0 |

involved in research ideation, experimental design, data analysis, or generation of technical content. All scientific contributions, methodology, and results are entirely the work of the authors.

