# OpenReview forum: "Hypergraph Neural Network for Integer Programming with High-Degree Terms"
_ICLR.cc/2026/Conference — ICLR 2026 Conference Withdrawn Submission_

### Official Review · Reviewer_9vcc · 2025-10-23

**Soundness:** 2
**Presentation:** 3
**Contribution:** 2
**Rating:** 2
**Confidence:** 4

**Summary:**

The paper propose to handle integer programs with high-degree terms by encoding each instance as a hypergraph and training a hypergraph neural network to predict variable values. After obtaining the prediction, a large-neighborhood search (with SCIP/Gurobi) is applied to maintain feasibility and improve objective.

**Strengths:**

The presentation is relatively clear and easy to follow.

**Weaknesses:**

1. Representing a monomial as a single hyper-edge with shallow message passing may not fully capture some characteristics (e.g., $x_1^2x_2^9$, where $x_2$ dominates). Without theoretical or empirical justification of expressivity and identifiability, it remains unclear whether the proposed architecture and encoding can aggregate sufficient signal, especially under a task of predicting solution values.
2. The novelty claim is weak. Representing higher-order terms with hyperedges is not new; for instance, prior work on QPs introduces auxiliary nodes for quadratic terms that play a role similar to hyperedges. Please clarify what is conceptually and technically different from other related works.

**Questions:**

Besides of those mentioned in Weakness section, there are some additional concerns on numerical results.
1. **SCIP vs. Gurobi in Table 1.** It’s surprising that Gurobi underperforms SCIP on the smallest and largest datasets, which is possibly due to aggressive presolve of Gurobi. Is it possible for the author to add a gap comparison plot (Gurobi vs. SCIP) and briefly describe the exact solver settings/presolve options used.
2. **Ablation parameter parity.** In the three ablation settings, do models have comparable parameter counts? If “Ours” uses substantially more neurons, it should outperform the remain two, which does not make much sense for the ablation.

---

### Official Review · Reviewer_wAXK · 2025-10-26

**Soundness:** 2
**Presentation:** 3
**Contribution:** 2
**Rating:** 2
**Confidence:** 4

**Summary:**

This paper proposes a learning-based framework for predicting solutions to integer programs with high-degree terms (IPHDs). The core contribution lies in introducing a hyper-graph representation for IPHDs and developing a corresponding GNN variant to process such hyper-graphs. However, I have concerns about a critical limitation: the proposed hyper-graph representation is not sufficiently expressive to distinguish between distinct IPHDs.

**Strengths:**

1. The paper is generally well-organized and presents a clear logical flow.
2. The idea of handling high-degree terms via hyper-edges is new to me.
3. Empirical results on two classes of IPHDs demonstrate some advantages of the proposed method over existing baselines.

**Weaknesses:**

1. **Critical Limitation**: The proposed hyper-graph representation is not bijective. In particular, there exist distinct IPHDs that share identical hyper-graph representations.
For example, consider IPHDs $\\{\min_{x} x_1|x_1x_2^2\leq 1,x_1^3x_2^1\leq 4\\}$ and $\\{\min_{x} x_1|x_1x_2^2\leq 4,x_1^3x_2^1\leq 1\\}$. These two problems yield the same hyper-graph representation. **If the representation itself cannot uniquely identify IPHD instances, it is unclear how subsequent GNN modules can reliably predict their solutions**. I suggest the authors verify this example and discuss this limitation explicitly.
2. **Unclear justification for using hyper-edges**: In my understanding, each hyper-edge could be equivalently replaced by a virtual node that connects the corresponding variable nodes, forming a tripartite graph with standard edges. Prior work, I remember, [1] has already adopted a similar tripartite construction for QCQPs. The authors may clarify the conceptual and practical advantages of their hyper-graph formulation over this alternative.

Minor comments and suggestions:
1. In Eq. (5), the definition of $h_v h_{ve}$ seems missing. Is it the Hadamard (element-wise) product of two vectors? Please clarify.
2. GAP-vs-time plot: Adding a figure that tracks the primal gap over time (corresponding to Table 1 or Table 2) would better illustrate the solving behaviors of different methods.
3. In Table 1, please avoid splitting the baseline “NeuralQP” into two lines for better readability.

---

[1] Wu, Chenyang, et al. "On representing convex quadratically constrained quadratic programs via graph neural networks." arXiv preprint arXiv:2411.13805 (2024).

**Questions:**

1. Is there a missing edge between x1 and c1 in Figure 1(b)?

---

### Official Review · Reviewer_qqv4 · 2025-10-30

**Soundness:** 3
**Presentation:** 2
**Contribution:** 3
**Rating:** 6
**Confidence:** 2

**Summary:**

This paper presents a hypergraph neural network approach for solving integer programming with high-degree terms, a challenging subclass of nonlinear integer programming. The method first constructs a high-degree term-aware hypergraph representation to capture both multivariable interactions within high-degree terms and variable-constraint interdependencies. It then designs an HNN architecture that integrates hyperedge-based and variable-constraint-based convolutions for accurate solution prediction, followed by a repair-and-refinement process to enhance results.

**Strengths:**

1. The paper effectively addresses a key limitation of traditional graph representations (e.g., bipartite graphs), which only capture pairwise interactions and fail to model the higher-order nonlinear structures inherent in IPHD. Its hypergraph-based framework fills this gap by explicitly encoding multivariable interactions in high-degree terms.

2. The HNN’s integration of two complementary convolution mechanisms—hyperedge-based convolution for high-order variable interactions and variable-constraint-based convolution for interdependencies between variables and constraints—is theoretically sound and practically effective, as validated by ablation studies.

**Weaknesses:**

1. Lack of Transparency on Data Collection Costs: Training the HNN requires a large volume of IPHD instances, which involves solving computationally expensive nonlinear integer programs. The authors do not report the time, computational resources (e.g., GPU/CPU hours), or scalability challenges encountered during training data generation.

2. Insufficient Analysis of Hypergraph Density: The paper does not discuss the density of the proposed hypergraph representation (e.g., number of hyperedges relative to variables/constraints) or how density impacts memory usage and inference speed. This is critical for evaluating the method’s scalability on extremely large or sparse IPHD instances.

3. Ambiguous Performance Assessment: The claim that “performance is not good” lacks specificity. While the method outperforms baselines overall, the authors should clarify if there are specific instance types (e.g., highly sparse, ultra-high-degree terms) or scales where performance degrades, and provide insights into the root causes.


4. Limited Support for Variable Problem Sizes: The current framework requires training on instances of similar scales to the test data (e.g., training on small/medium scales for large-scale testing). It does not address whether a single model can handle variable problem sizes directly, which is a key practical requirement for real-world optimization tasks.

**Questions:**

Please see Weakness.

---

### Official Review · Reviewer_Ro3M · 2025-11-01

**Soundness:** 3
**Presentation:** 2
**Contribution:** 2
**Rating:** 4
**Confidence:** 2

**Summary:**

This paper introduces a hypergraph neural network (HNN) framework for solving integer programming problems with high-degree terms (IPHD). The authors propose a hypergraph representation that captures multi-variable interactions in high-degree terms via hyperedges and variable-constraint relationships via standard edges. The HNN architecture integrates hyperedge-based convolution for higher-order information and variable-constraint-based convolution for interdependencies, predicting solution values that are refined through a neighborhood search process using off-the-shelf solvers.

**Strengths:**

The paper's key strength lies in its generalization beyond quadratic programming to arbitrary high-degree terms, addressing a gap in learning-based methods for nonlinear integer programming. The hypergraph representation effectively models complex interactions. The reported empirical results are relatively compelling, though I am not very confident in this since I am mainly a theorist.

**Weaknesses:**

The idea of hypergraph representation is already proposed in Appendix F of [1]. So I am not sure about how novel this paper is. I think the authors should at least state the proposed ideas in [1].

[1] Ziang Chen, Xiaohan Chen, Jialin Liu, Xinshang Wang, and Wotao Yin. Expressive power of graph neural networks for (mixed-integer) quadratic programs. In the Forty-second International Conference on Machine Learning, 2025

**Questions:**

None.

---

### Official Review · Reviewer_rCgx · 2025-11-01

**Soundness:** 3
**Presentation:** 3
**Contribution:** 2
**Rating:** 4
**Confidence:** 5

**Summary:**

The paper proposes a hypergraph neural network (HNN) for integer programming with high-degree terms (IPHD). It represents each higher-order monomial as a hyperedge, uses two message-passing routes—hyperedge-based (variables ↔ high-degree terms) and variable-constraint-based (variables ↔ constraints)—to predict variable values, and then applies Q-repair plus iterated multi-neighborhood search with Gurobi/SCIP for repair and improvement. Experiments on quadratic (QMKP, RandQCP), QPLIB, and quintic CFLPTC show competitive or better gap% than baselines, including NeuralQP and pure solvers, with strongest gains on quintic tasks.

**Strengths:**

- Important problem: bringing learning-based methods to IP with high-degree terms is valuable. The empirical results are solid, especially on quintic instances where direct handling avoids quadratic reformulation overheads. The framework integrates smoothly with standard solvers and demonstrates scalability.

**Weaknesses:**

- Novelty vs. NeuralQP is unclear. NeuralQP already uses a hypergraph neural network and the same Q-repair + iterated multi-neighborhood refinement pipeline. The main additional component here appears to be adding explicit variable–constraint edges (E) and generalizing hyperedge features to carry degree information. It is not convincingly shown that NeuralQP cannot be readily extended to >2-degree by augmenting its hyperedge construction and features; thus the core contribution risks being seen as “NeuralQP with extra edges.”
- Attribution and ablation are insufficient to isolate what really drives the gains beyond NeuralQP.
- Theoretical justification is limited. There is no analysis that the proposed variable–constraint edges (E) or other design is necessary/sufficient for higher-order expressivity vs. a straightforward extension of NeuralQP.

**Questions:**

- Can the authors implement and report a NeuralQP-HP (higher-order) variant that (a) generalizes its hyperedges to arbitrary-degree terms with degree/coeff features and (b) adds variable–constraint edges, keeping the rest intact? This would directly test whether the proposed architecture brings benefits beyond a minimal NeuralQP extension.

---

### Note · Authors · 2025-12-11

**Comment:**

We sincerely acknowledge the comments and efforts from all reviewers and chairs.

**Withdrawal Confirmation:**

I have read and agree with the venue's withdrawal policy on behalf of myself and my co-authors.